# Growth and Drug Interaction Monitoring of NIH 3T3 Cells by Image Analysis and Capacitive Biosensor

**DOI:** 10.3390/mi12101248

**Published:** 2021-10-14

**Authors:** Gayoung Lee, Jaehun Jeong, Yeeun Kim, Dahyun Kang, Sooyong Shin, Jongwon Lee, Sung Ho Jeon, Moongyu Jang

**Affiliations:** 1School of Nano Convergence Technology, Hallym University, Chuncheon 24252, Korea; ygy2221@daum.net (G.L.); jaehun952@naver.com (J.J.); dms1334@naver.com (Y.K.); audrii@naver.com (D.K.); 2Department of Life Science and Multidisciplinary Institute, Hallym University, Chuncheon 24252, Korea; ccyzzo@hallym.ac.kr (S.S.); m20020@hallym.ac.kr (J.L.); sjeon@hallym.ac.kr (S.H.J.); 3Center of Nano Convergence Technology, Hallym University, Chuncheon 24252, Korea

**Keywords:** capacitive biosensor, NIH 3T3 cell, cell growth, cell–drug interaction, real-time monitoring

## Abstract

Capacitive biosensors are manufactured on glass slides using the semiconductor process to monitor cell growth and cell–drug interactions in real time. Capacitance signals are continuously monitored for each 10 min interval during a 48 h period, with the variations of frequency from 1 kHz to 1 MHz. The capacitance values showed a gradual increase with the increase in NIH 3T3 cell numbers. After 48 h of growth, 6.67 μg/mL puromycin is injected for the monitoring of the cell–drug interaction. The capacitance values rapidly increased during a period of about 10 h, reflecting the rapid increase in the cell numbers. In this study, we monitored the state of cells and the cell–drug interactions using the developed capacitive biosensor. Additionally, we monitored the state of cell behavior using a JuLi^TM^ Br&FL microscope. The monitoring of cell state by means of a capacitive biosensor is more sensitive than confluence measuring using a JuLi^TM^ Br&FL microscope image. The developed capacitive biosensor could be applied in a wide range of bio-medical areas; for example, non-destructive real-time cell growth and cell–drug interaction monitoring.

## 1. Introduction

There is a growing interest and need for electrochemical nano-sized device technology that functions better than conventional electronic devices by utilizing the phenomena and functions of life. Accordingly, biosensor technology is a device technology that combines various technological elements such as informational technology (IT), nanotechnology (NT), and molecular biology technology (BT), and is expanding its scope by being applied to various fields. Among these fields, studies on biological signal monitoring are actively underway in medical fields such as diagnosis, treatment and home healthcare [1,2,3].

Frequently, the biological method of observing the conditions of cells has been performed by dyeing cells and observing them directly under a microscope. This method requires the dyeing of cells (pre-treated) and monitoring under a microscope to check the cells’ condition at regular intervals. However, cell dyeing is a destructive method and affects the status of cells. Therefore, the same cells cannot be continuously observed, and multiple samples must be grown to observe the cells, which causes errors resulting from the samples. Thus, biological analysis methods have the disadvantage of being time-consuming and complex, resulting in more errors. However, the capacitive biosensors used in this study have the advantage of being simpler, because they do not need any extra treatment processes to measure electrical signals, allowing the cells to be monitored continuously in real time. Therefore, continuous research is being conducted to measure cell state using an electrical approach [4,5,6].

Capacitive biosensors can monitor the condition of cells using the electric cell-substrate impedance sensing (ECIS) technology. The ECIS method uses the insulating characteristics of cells to monitor the condition of cells by looking at one cell as a capacitor and increasing the impedance through contact with the electrode and the covered area of the electrode. The impedance of cells combined with electrodes is primarily determined by the three-dimensional geometry of the cells. When a cell changes in shape, the path of the current through the cell body and its surrounding path change, resulting in a corresponding increase in impedance [7,8,9,10]. Therefore, measuring the time-decomposing impedance allows real-time tracking of cell shape changes and can be used for bioanalysis purposes.

Since these animal cell shapes respond very sensitively to changes in metabolism as well as chemical, biological, and physical stimuli, ECIS methods can be applied in various experimental environments in the cell biological laboratory. ECIS technology can be used in sensors for cytotoxic research, drug development, and as a noninvasive tool for the tracking of cell adhesion on the surface of the body.

Therefore, in this study, a sensor with a platinum electrode with a specific pattern was manufactured for measurement based on ECIS analysis, and cells were grown on these sensors to measure the state of the cells by applying a constant alternating current.

## 2. Materials and Methods

The manufacturing processes of the sensor is shown in Figure 1. Sensors are manufactured on glass slides to allow cells to be observed by a microscope. First, the glass slide was cleaned, and then the process was carried out to produce capacitive biosensor patterns on top. At this time, the photolithography process was carried out for a more precise pattern process. The photoresist was coated with a spin coater at 5000 rpm for about 35 s and baked on a hot plate. Then, the Mask Alignment System (EVG 610) was used to align the pattern mask and exposed for 8.3 s. After that, the AZ 300 MIF developer was used to develop the sample and remove the photoresist of the pattern. RF/DC magnetron sputter equipment was used to deposit electrode patterns. In order to facilitate adhesion between the slide glass and Pt patterns, 5 nm of Cr was deposited and 80 nm of Pt was deposited. After the deposition was completed, the lift off process was carried out on the sample to leave only the pattern. Annealing was performed to make the electrode connection stronger, and the PDMS well was attached over the capacitive biosensor pattern to create a space for the cells to grow.

First, the inside of the well was sterilized with 70% Ethanol (EtOH) and was coated with poly-L-lysine for 40 min to facilitate cell adhesion. The inside of the poly-L-lysine-coated well was dried overnight. At this time, the inside of the well was irradiated with UV for 40 min and was sterilized completely. The cells were seeded on a completely dried sensor. The number of seeded cells was adjusted to 10,000, and as the seeded cells sank, they stuck to the bottom inside of the well, and grew over time. As these cells grew, they were monitored by sensors to check the condition of the cells.

The cells used were NIH 3T3 cells, which are fibroblasts extracted from mouse embryos. The measurements made with sensors were as follows. After seeding the cells, six samples were incubated in the CO_2_ incubator for 48 h at a time to measure capacitance according to changes in the number of cells within the sensor. After 48 h of cell culture, cells were fully filled inside the well. When the drug was injected, 100 μL of the existing medium in the well was removed. Then, 200 μL of puromycin (Gibco, New York, NY, USA, A11138-03) diluted in the medium at a concentration of 10 μg/mL was added to inject the drug with a concentration of 6.67 μg/mL. The cell–drug interaction processes were monitored for 48 h.

## 3. Results and Discussion

Using biosensors manufactured using the above method, the capacitance of cells was measured as a function of frequency and the capacitance of cells at a particular frequency. Measurements were made using Keithley 4200-SCS, Agilent 4284A, and Keithley 707B Switching Matrix, and microscopic images were observed using NanoEnTek′s live cell movie analyzer (JuLI^TM^ Br & FL, Waltham, MA, USA). The measuring process is shown in Figure 2.

The measurement principles of sensors used in this study are as follows. ECIS technology is widely studied in the field of biosensors for monitoring cell states in real time as an electrical approach to monitor cell responses. A constant small AC current is applied between electrodes and the potential is measured, in which the outer and inner fluids of the cell are conductors and the cell membrane is an insulator with a phospholipid double layer. When there is a voltage difference across the cell membrane, electric charges accumulate at the interface because the current cannot flow directly through the insulator, and usually, in low-frequency regions, the electric field polarizes the semi-ionic cloud, resulting in an electrical dipole. Thus, it can be seen that the capacitance increases as the cell grows, creating a stronger electrical dipole by the cell membrane grows [11,12]. The insulation properties of the cell membrane create resistance to the current flow, which increases the potential between electrodes. The ECIS analysis method has the advantage of being able to monitor cell status in real time using the in vitro method, which is easy to use and non-invasive [13,14].

Puromycin, a drug used to kill cells, is a type of antibiotic that functions as a protein synthesis inhibitor. It has a similar structure to the end of the aminoacyl tRNA. For protein synthesis, aminoacyl tRNA is produced during the activation step. Where aminoacyl tRNA is a preparation for peptide binding of amino acids. The presence of puromycin in place of aminoacyl tRNA in the protein synthesis process inhibits protein synthesis by premature termination of the polypeptide chain, which makes it impossible for cells to survive because they cannot make proteins normally [15,16].

Puromycin is also a drug that causes apoptosis [17,18]. Looking at the mechanisms of the cells when apoptosis occurs, when puromycin is injected into the cells, the cells react with drugs for a certain period of time, resulting in various interior reaction before the cell death [19]. By measuring cells using the above principles, the capacitance of the cells according to frequency sweep was identified, and the frequency at which NIH 3T3 cells most sensitively represented cell changes was 367 to 440 kHz. The results are shown in Figure 3.

As a result, when checking the capacitance of cells, the most reliable frequency is determined as 400 kHz, and we checked the results together up to 300 and 500 kHz, simultaneously. Figure 4 shows the measured results of 300, 400 and 500 kHz, respectively. The capacitance–time graph shows that the capacitance value increases from 0 to 48 h. After the drug’s injection into the well, the capacitance value decreases after about 7–8 h. The red colored data in the graphs in Figure 4a–c show media-only capacitance data for comparison, and there are no remarkable changes in media-only cases. These results were compared with microscopic images from cell growth and death cases. The graph in Figure 4d shows the reproducibility of the experiment. In the four samples shown in this graph, differences in capacitance values appear as cell growth and death differ for each sample. However, it can be seen that the overall shape of the graph is consistent.

Figure 5 shows the comparison between images observed by the JuLI^TM^ Br&FL microscope and measured electrical signals. In the cell growth stage, about 2 or 3 h after cell seeding, all of the cells sink down on the bottom. Then, cells are attached on the bottom and grow. When the cells were seeded, about 19,894 cells/cm^2^ were contained in the well of the sensor, and about 79,577 cells/cm^2^ were contained after the cells were fully grown. At electrical signal in the graph, the capacitance value increases from 0 to 48 h. Therefore, the image in the growth stage shows that the cells are increasing over time, and the capacitance value is also increasing. When the cells were fully filled in the well, the drug was injected. However, there are no significant changes in the microscope images during the eight hours from the drug injection time in the well, and the rising graph can be seen in electrical signals. The part mentioned earlier is currently under study and is thought to be caused by the mechanism inside the cell when cells and drugs react. From that point, capacitance can be seen to decrease rapidly from 8 to 20 h after drug injection. The microscope image shows that the cells die after 8 to 20 h after drug injection too. After 24 h, we can confirm that all the cells are dead and there is no change in the electrical signal.

## 4. Conclusions

A capacitive biosensor was manufactured for NIH 3T3 cell growth and cell–drug interaction monitoring. We established an NIH 3T3 cell culture and drug reaction with steady conditions on the sensor. Cell growth was measured for 48 h after seeding about 10,000 cells, and cell state and death following drug injection were observed. When comparing the results between signals from sensors with real cells, the time-dependent live and dead electrical signal showed similar tendencies as optical microscope image results. At 8 h after puromycin injection, capacitance values decreased sharply, reflecting the death of cells. Cell measurements using capacitive biosensors confirmed that capacitance had different results depending on the frequency. In the C (t)/C (0) and frequency graph, capacitance value variations were most sensitive from 367 to 440 kHz. Biosensors using the existing ECIS method measured cell conditions in a bulky-sized pattern. However, capacitive biosensors can detect cell changes more sensitively by using fine patterns in small areas. In this study, we monitored the state of various cells and the cell–drug interactions using the developed capacitive biosensor chip by using electric cell-substrate impedance sensing (ECIS) technique.

## Figures and Tables

**Figure 1 micromachines-12-01248-f001:**
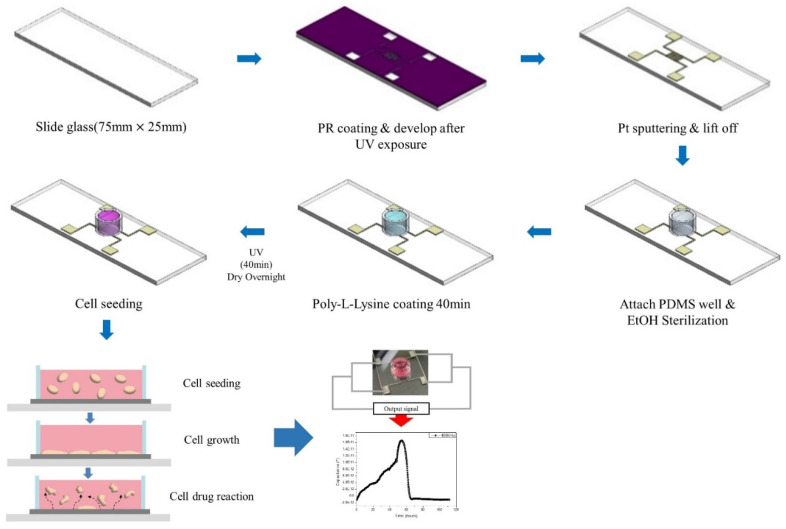
Capacitive biosensor manufacturing process.

**Figure 2 micromachines-12-01248-f002:**
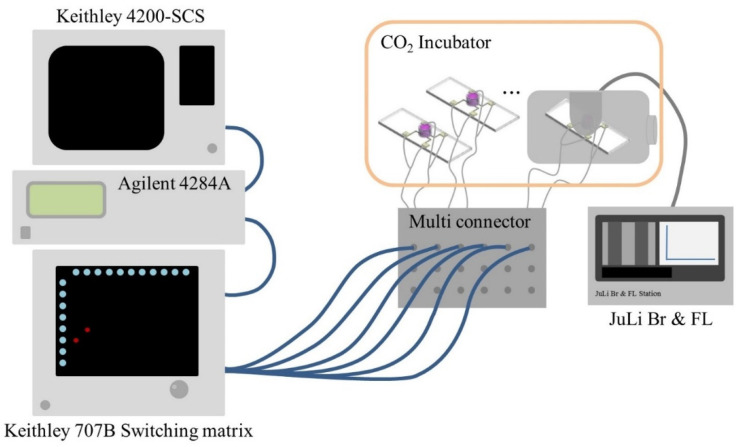
Cell measuring process on the biosensor.

**Figure 3 micromachines-12-01248-f003:**
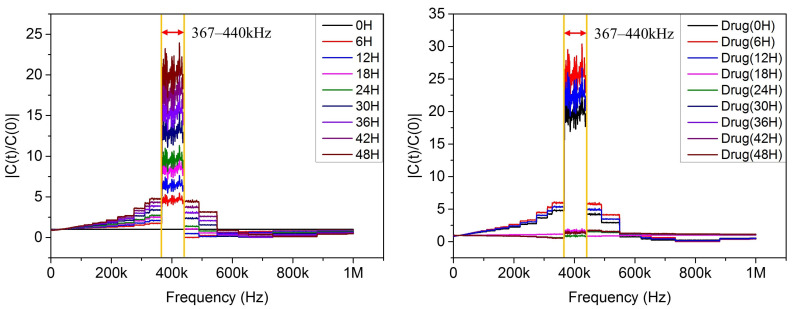
The graph of measuring capacitance by frequency sweep on cell growth (**left**) and death (**right**) state.

**Figure 4 micromachines-12-01248-f004:**
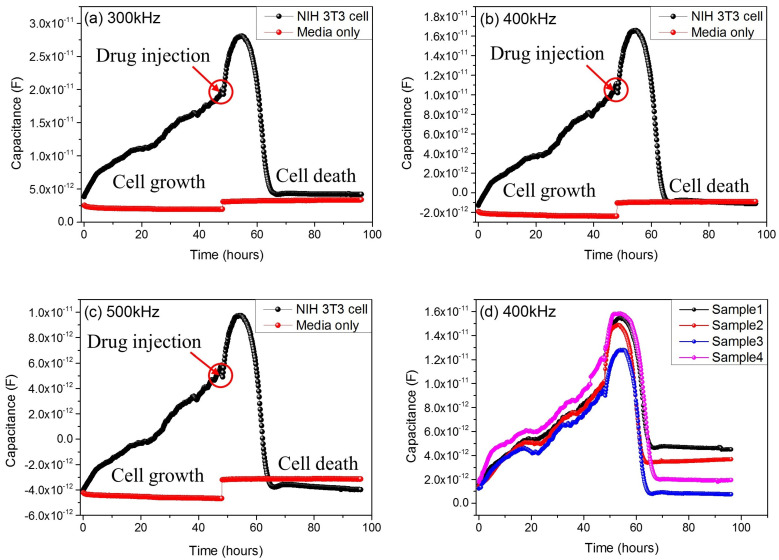
The graph of capacitance to time flow in sensitive frequency band (300 kHz (**a**), 400 kHz (**b**), 500 kHz (**c**)) and multiple plots of four different samples at 400 kHz for the reproducibility confirmation (**d**).

**Figure 5 micromachines-12-01248-f005:**
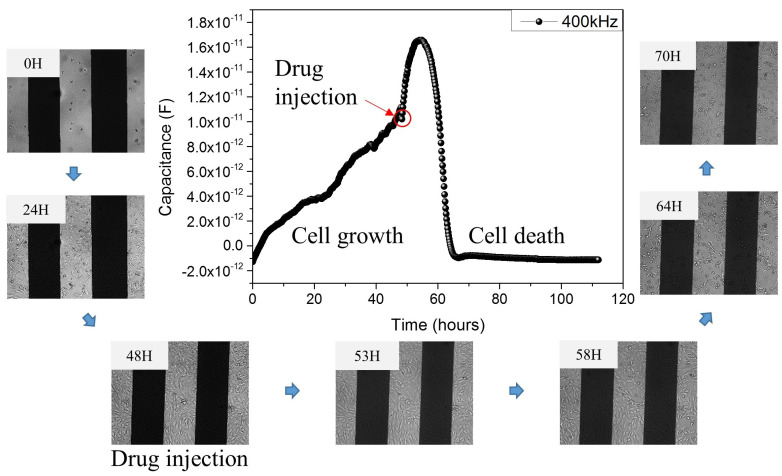
Comparison of microscope image and electrical signal on cell growth and death state.

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
