# Peer review of "Growth and Drug Interaction Monitoring of NIH 3T3 Cells by Image Analysis and Capacitive Biosensor"

_micromachines, 2021, doi:10.3390/mi12101248_

Round 1

Reviewer 1 Report

The authors of this paper used the developed biosensor to monitor cell status and cellular drug response. The performance of the biosensor was compared with the cell state observed by the microscope. The developed biosensor has the potential to be applied to real-time cell growth and cell drug response monitoring.

I believe the work described is original and therefore suitable for publication in Micromachines. However, before accepting this article for publication, I have several points that the authors need to address:

  • In Figure 5, the authors show the cell microscope image and cell capacitance test results at different time periods. We hope the authors can quantify the cell images taken under the microscope, for example, the number of cells per unit area. Then the quantified results can be compared with those of cell capacitance test.
  • In this article, the authors mainly measured the relationship between capacitance and frequency in different cell states. Impedance mainly refers to the vector sum of resistance and reactance. This article uses the name of the impedance biosensor, which seems inappropriate. It is recommended to use a capacitive sensor instead.
  • The authors require consistency in using the lower and upper case of the initial letter, like “Informational Technology” in line 30 and “molecular Biology technology” in line 31, “Electrical Cell-substrate Impedance Sensing” in line 175 and 46-47. The authors also need to avoid grammar mistake like “an electrical approaches” in line 46, the word ‘approach’ should use the singular form.

Author Response

Dear. Reviewer 1

We greatly appreciate for the valuable comments to the manuscript, entitled “Growth and drug reaction monitoring of NIH 3T3 cells by image analysis and capacitive biosensor”.

All the issued items are revised according to the reviewer’s comments.

The revised parts are highlighted as blue color for the comvenience.

Best regards,

Moongyu Jang.

Reviewer 2 Report

The authors monitored the growth and drug reactions of cells using a biosensor which is made of an image analysis instrument and impedance detecting apparatus. In general, this manuscript lacks insightful perspectives, and experiments in this manuscript were not well-designed. I am unable to recommend this manuscript in its current form due to the following detailed reasons.

  1. There are numerous papers on ECIS, but the advantages and differences in this work were not highlighted.
  2. It seems that the growth and drug reactions of cells were monitored only once. The repeatability of this experiment was not guaranteed.
  3. This experiment lacks a control group. The well without any cells should be measured and monitored in order to calibrate the whole set of instruments.
  4. I recommend that Figure 3 should be re-organized because too many points overlap and details cannot be seen clearly. Three graphs in Figure 4 can be drawn in one graph.
  5. In figure 5, for the both pictures at left bottom (before and after drug injection at 48H), they are totally different in texture that made of cells. That makes readers doubt that both pictures are from different experiments.

Author Response

Dear. Reviewer 2

We greatly appreciate for the valuable comments to the manuscript, entitled “Growth and drug reaction monitoring of NIH 3T3 cells by image analysis and capacitive biosensor”.

All the issued items are revised according to the reviewer’s comments.

The revised parts are highlighted as blue color for the comvenience.

Best regards,

Moongyu Jang.

Round 2

Reviewer 2 Report

Comments addressed.